# Magnetic Fields and Cancer: Epidemiology, Cellular Biology, and Theranostics

**DOI:** 10.3390/ijms23031339

**Published:** 2022-01-25

**Authors:** Massimo E. Maffei

**Affiliations:** Department Life Sciences and Systems Biology, University of Turin, Via Quarello 15/a, 10135 Turin, Italy; massimo.maffei@unito.it; Tel.: +39-011-670-5967

**Keywords:** magnetic field, cancer, epidemiology, therapy, diagnostics, theranostic, MRI, magnetic nanoparticles, nanomedicine, reactive oxygen species

## Abstract

Humans are exposed to a complex mix of man-made electric and magnetic fields (MFs) at many different frequencies, at home and at work. Epidemiological studies indicate that there is a positive relationship between residential/domestic and occupational exposure to extremely low frequency electromagnetic fields and some types of cancer, although some other studies indicate no relationship. In this review, after an introduction on the MF definition and a description of natural/anthropogenic sources, the epidemiology of residential/domestic and occupational exposure to MFs and cancer is reviewed, with reference to leukemia, brain, and breast cancer. The in vivo and in vitro effects of MFs on cancer are reviewed considering both human and animal cells, with particular reference to the involvement of reactive oxygen species (ROS). MF application on cancer diagnostic and therapy (theranostic) are also reviewed by describing the use of different magnetic resonance imaging (MRI) applications for the detection of several cancers. Finally, the use of magnetic nanoparticles is described in terms of treatment of cancer by nanomedical applications for the precise delivery of anticancer drugs, nanosurgery by magnetomechanic methods, and selective killing of cancer cells by magnetic hyperthermia. The supplementary tables provide quantitative data and methodologies in epidemiological and cell biology studies. Although scientists do not generally agree that there is a cause-effect relationship between exposure to MF and cancer, MFs might not be the direct cause of cancer but may contribute to produce ROS and generate oxidative stress, which could trigger or enhance the expression of oncogenes.

## 1. Introduction

Public concern about electromagnetic fields (EMFs) from power systems is increasing along with the electricity demand, wireless technologies, and changes in work systems and social behavior [1,2,3,4]. For modern populations, extremely low-frequency (ELF) electric and magnetic fields (MFs) are common exposures and complex biological mechanisms underly the potential effects of externally-applied MFs [5,6]. In 2002, the International Agency for Research on Cancer (IARC) categorized ELF (including the power frequencies of 50 and 60 Hz) MFs as “possibly carcinogenic to humans” [7].

Controversial and often contradictory scientific reports continue to stimulate debates on the biological effects of EMFs, often leading to confusion and distraction which hamper the development of univocal conclusions on the real hazards that are caused by EMFs [8]. In this review the association between MF and cancer will be reviewed by considering the effect of MF in causing cancer as well as the application of MF as a therapeutic and diagnostic (theranostic) tool. Epidemiological studies, including both domestic/residential and occupational data, as well as human and animal cell studies that were published in the last 20 years will be also considered to provide an overview of the state of the art literature.

The strategy that was implemented to carry out this review was based on a deep search in the databases Web of Science (2000–2021), PubMed (2000–2021), and the EMF Portal (https://www.EMF-portal.org/en, accessed on 1 December 2021). By considering as entries the terms “cancer” AND “magnetic field” the total number of Web of Science Core Collection papers in the period from January 2000 to December 2021 was 12,364, whereas, for the same period, the total number of papers in PubMed was 11,539. The selection of papers was done on the terms: diagnostics, therapy, epidemiology, policy, along with a selection of cancer types, and the exclusion criteria was the impossibility to obtain a full text or the lack of specificity with the selected areas of the review.

Despite the narrative nature of this view, quantitative data on MF exposure and methodologies are described in Appendix A (Appendix A), whereas a Appendix A (Appendix A) contains all references that were cited in this article in addition to many other references.

### 1.1. Definition and Natural/Anthropogenic Sources of Magnetic Fields

EMFs are present everywhere in our environment. Electric fields are produced by the local build-up of electric charges in the atmosphere that are associated with thunderstorms. The Earth’s MF, or geomagnetic field (GMF), is the principal source of static fields (SFs) [9]. It interacts with the geosphere and the biosphere and plays a major role in shielding the harmful effects of cosmic radiation. Different areas inside our planet are responsible for the GMF which can be represented as the sum of MFs of several sources:FΤ = F_0_ + F_m_ + F_a_ + F_e_ + δF
where F_0_ is the dipolar component of the GMF, F_m_ is the field of world anomalies that are associated with the heterogeneity of the planet interior (non-dipolar field), F_a_ is the magnetization of rocks in the Earth’s crust (anomalous field), F_e_ is the external sources field, and *δ*F is the field variation that is also associated to external causes. The main GMF is also represented by the sum of the dipolar and non-dipolar fields (F = F_0_ + F_m_).

The GMF is composed of three orthogonal vectors: *X*, *Y,* and *Z*. The combination of the two horizontal vectors yields the horizontal component *H*, which is aligned in the direction of the compass needle and that can be expressed as:$$H=\sqrt{X^{2}+Y^{2}}$$

Whereas the total field intensity, which at the poles is directed towards the center of the planet, can be expressed as:$$H=\sqrt{X^{2}+Y^{2}+Z^{2}}$$

The angle that is formed between *H* and the geographic north is the declination, *D*, whereas the inclination, *I*, is the angle between the horizontal plane and the vector of total field intensity *F*. The international SI system the magnetic induction or magnetic flux density (*B*) is measured in Tesla (T) and its subunits (µT = 10^−6^ T; nT = 10^−9^ T). One tesla equals one Weber per square meter, corresponding to 10^4^ gauss (G), which is the unit of magnetic field in the centimeter-gram-second system. Thus, 1 G = 100 µT. 

The magnetic flux density, *B,* is linked to the magnetic field strength, *H,* by a material constant, the magnetic permeability *µ* (also called magnetic conductivity).
*B* = *µ* × *H*

The magnetic permeability, *µ*, is a measure of the permeability of materials for MFs.

The power flux density, *S*, of the EMF consists of energy fractions of the electric and MF components and is measured in Watts per square meter (W m^−2^). The field strength decreases with increasing distance from the field source.

The strength of the GMF at the surface of the Earth ranges from over 60 μT around the magnetic poles in northern Canada, the south of Australia, and in parts of Siberia to less than 30 μT in an area that includes most of South America and South Africa (the so-called South Atlantic anomaly) [10]. In the Earth’s history, the GMF has changed with the so-called geomagnetic reversals, where the GMF was characterized by periods (more-or-less extended) with the same polarity. These reversals occurred some hundred times since the Earth’s formation, with intervals between the polarity phases estimated around 300,000 years. The present normal polarity started around 780,000 years ago; therefore, an imminent geomagnetic reversal would not be so unexpected. The South Atlantic anomaly, a zone with significant reduction of the GMF intensity that is located in front of Brazil/Argentina, could be the initial symptom of a future change of polarity [2]. Changes in GMF intensity imply a reduction of the GMF shield against cosmic radiation, with possible consequences for all living organisms, which cannot avoid the effects of the GMF [11]. 

It is not clear whether the GMF can contribute to potential health risks, being present in our planet before the evolution of living organisms; however, one question remains whether different values of GMF in different countries in which epidemiological studies of anthropogenic sources of MF have been performed might affect the results of those studies [12]. 

Besides natural sources, the EM spectrum also includes fields that are generated by human-made sources. For instance, X-rays are generated and used for diagnosis, power sockets are associated with low frequency electromagnetic fields (LF EMFs), and various kinds of higher frequency radio waves are used to transmit information—whether via TV antennas, radio stations, or mobile phone base stations. 

Static magnetic fields (SMF) (with direct current, DC) or alternating magnetic fields (AMF) (with alternate current, AC) are formed, depending on the current feed. The polarity of AMF changes according to the cyclic changes in the direction of the current flow (e.g., 100 polarity changes per second with 50 Hz AC), whereas in SMFs, the polarity is unchanged.

The scientifically documented interaction with the organism allows the classification of non-ionizing electromagnetic fields (NI EMF) into low frequency (LF) and radio frequency (RF). The stimulation or excitation of nerves, muscles, and sensory receptors may occur below a threshold of 1 MHz; however, values that are higher than 1 MHz generate only thermal effects. EMFs and radiation cover a wide frequency range. The NI radiation range of the EM spectrum up to 300 GHz comprises SMF (0 Hz) and LF fields (≤300 Hz), the intermediate-frequency range between approximately 300 Hz and 10 MHz, and the RF range from 10 MHz to 300 GHz.

Table 1 summarizes the classification of MFs based on type of radiation, field, frequency, and wavelength along with some examples and general effects. Static, non-ionizing electric, and MFs that occur as a by-product. SF (0 Hz) occurs in batteries, at high voltage direct current transmission lines (HVDC lines), where underground cables are present, with permanent magnets, between objects with different electrical charges, and in the GMF. In general, the kind of the MF and the level of the magnetic flux density correspond to those of the GMF. Inside of the converter stations, SMFs occur, and their strengths depend on the voltage and the amount of flowing current. In medicine, strong SMF are used in magnetic resonance imaging (MRI, see also Section 4.1). During an MRI procedure the patient is exposed a strong SMF normally from 1.5–3 T. Only in research facilities, magnetic fields from 7 T up to 9.4 T are used. A huge diversity of products with magnets that are in close proximity to their surface in pillows, belts, bracelets, blankets, pendants, patches, or insoles exhibit SMF in the range between 0.03 and 0.3 T. However, the levels are reduced to a tenth at a 3–4 mm distance from the surface. Therefore, at a distance of several centimeters, the magnetic flux density lowers down to that of the natural GMF.

LF refers to the frequency range 0–100 kHz. The energy of the EMF that is absorbed in biological tissue and is converted into heat defines the specific absorption rate (SAR) that is obtained by exposure to a frequency that is between 100 kHz and 10 GHz. The SAR is expressed in Watts per kilogram of tissue (W kg^−1^) based on an average exposure time of six minute intervals, during which a balance between the energy input and the heat dissipation in the tissue is reached. It is possible to distinguish between the exposure of the whole body or parts of the body by averaging over different body masses. All electrical applications that are run on power supply (railways, electrical appliances in the home, and at working places) lie in the range of LF AMF up to 1 kHz (wavelengths larger than 300 km) (low frequency (0.1 Hz–1 kHz)). No extremely LF AF occur in nature. ELF AFs are generated by technical appliances such as power lines, wiring, and household appliances. ELF Electric and MFs are generated by the power lines and their strength and distribution in the area surrounding the power lines depend on several parameters (including voltage, amperage, tower shape, as well as alignment, and number, and slackness of the lines). The strength of electric field is mainly found beneath the power lines; however, this effect rapidly diminishes with increasing distance from the power line [13]. Electric cars are a significant source of very high MFs due to the electric motor and large batteries, especially during starting and stopping. In electric cars ELF MF dominate, but intermediate frequency fields can occur also [14].

In the natural environment EMFs with intermediate frequencies (1 kHz–10 MHz) can be generated during the so-called sferics, which are broadband EM impulses that occur in the Earth’s atmosphere as a consequences of lightning discharges. Sferics may extend from a few kHz to several tens of kHz (3–100 kHz) [15]. Intermediate frequency includes the lower range of the radiofrequency band with its corresponding applications, but also applications that are working with specific frequencies, such as induction cookers and electronic article surveillance systems in stores, as well as many industrial and medical applications. 

Radio frequency (30 kHz–300 GHz) includes a range of "broadcasting frequencies" (between 30 kHz and 300 MHz; wavelengths from 10 km to 1 m) covering long wave radio broadcasting, amplitude modulation (AM) radio broadcasting, shortwave radio broadcasting, and frequency modulation (FM) radio broadcasting (authorized in the very high frequency range). Terahertz waves are also in the non-ionizing radiation spectral range, between 300 GHz and 10 THz (wavelengths from 1 mm to 30 µm). For example, they are used for quality control of industrial products, at some airports in body scanners for security control, or in skin cancer scanning systems [16]. 

The following range of 384 THz to 789 THz (780 nm to 380 nm) is referred to as visible light. This is succeeded by the ranges of ultraviolet radiation and the ionizing radiation with even shorter wavelengths. 

As noted above, the magnetic field strength around a conductor increases with rising electric current strength and decreases with growing distance from the field source. It is dependent on the type of source how fast the field decreases (Figure 1).

### 1.2. Public Health Initiatives and Concern

In 1996 the World Health Organization (WHO) launched a large, multidisciplinary research effort to respond to growing public health concerns about the possible health effects from exposure to EMF sources. The International EMF Project, open to any WHO Member State government, brings together current knowledge and available resources of key international and national agencies and scientific institutions. Among the aims and scopes of the EMF Project are: (a) develop and publish a health risk assessment on EM RF fields; (b) develop and disseminate information materials on risk management policies of EMF; (c) provide technical support to national authorities and international organizations regarding NI radiation; (d) establish an inter-agency committee on NI radiation safety to exchange information and harmonize activities; and (e) develop international standards for protection against NI radiation [17].

Worldwide, many countries set their own national standards for exposure based on the guidelines that are set by the International Commission on Non-Ionizing Radiation Protection (ICNIRP), a non-governmental organization that was formally recognized by WHO. Risk assessment analyses that are based on publicly available data are used to help formulate government guidance on occupational MF by also considering the cancer cases that were prevented and the monetary benefits accruing to society by reducing workplace exposures [18]. An overview of the current knowledge regarding EMF-related health risks including recommendations for the diagnosis, treatment, and accessibility measures of electromagnetic hypersensitivity (EHS) to improve and restore individual health outcomes as well as for the development of strategies for prevention has been recently published [19]. The International Radiation Protection Association (IRPA) represents national radiation protection societies [20]. An updated and reliable source of information is provided by the EMF Portal (https://www.EMF-portal.org/en, accessed on 30 December 2021).

In general, the type and extent of the cautionary policy that is chosen critically depends on the strength of the evidence for a health risk and the scale and nature of the potential consequences. In many countries, the adoption of a principle of caution or prudent avoidance implies the low-cost avoidance of unnecessary exposure as long as there is scientific uncertainty about its health effects [21,22]. However, still some policies are effective in preventing new situations with long-term exposure of children to MFs from overhead power lines, but these generally do not include underground cables and other sources of MFs [23]. Preventive measures and precautionary principles are necessary to warrant the reduction of exposure to children because of their greater sensitivity to ELF EMF [24,25,26]. The American Academy of Pediatrics set out new recommendations to decrease the adverse effects of exposure on children also to mobile phones [27].

## 2. Epidemiological Studies Evaluating MF and Cancer Relationships

It is known that epidemiological studies alone cannot be used to determine a clear cause and effect relationship when considering MF and cancer. This is mainly because epidemiological studies evaluate only the statistical associations between exposure and disease, which may not be necessarily caused by the exposure. Only the presence of a consistent and strong association between the exposure and the effect, a clear dose-response relationship, support that is provided by relevant animal studies, a credible biological explanation, and above all, if there is consistency between the studies can support cause and effect conclusions. In studies involving EMF and cancer most of these factors are generally missing. Studies of the potential health effects of EMF have concentrated on the MF because it is generally assumed to be the component that is most likely to have biological effects [28]. Since the first evidence determining the relationship between the ELF EMF and leukemia in children [29], epidemiological studies in this context increased and the IARC classified the ELF EMF in group 2B, a “possible carcinogen” to humans, whereas static electric and MFs are not classifiable as carcinogenic to humans (Group 3) [30]. Although it is generally accepted that EMFs can exert biological effects, in general, epidemiological studies show a weak and sometimes inconsistent association between exposure to power-frequency fields (PFF) and cancer. In most cases, the studies fail to show a dose-response relationship [31,32]. The opposite happens in laboratory studies where PFF points towards causing or contributing to cancer (see below). The application of “Hill’s criteria” (i.e., strength, plausibility, specificity, biologic gradient, consistency, coherence, experimental evidence, temporality, and analogy) to laboratory and epidemiological studies shows a weak evidence for a causal association between cancer and the exposure to PFF [33].

Because cancer is one of the significant problems of global health, epidemiologic studies have faced the question of whether occupational and residential exposure to ELF EMF might be carcinogenic. There are three main explanatory hypotheses that appear in the literature: (i) the EM hypothesis, attributing EHS to EMF exposure; (ii) the cognitive hypothesis, assuming that EHS results from false beliefs in EMF harmfulness; (iii) the attributive hypothesis, considering EHS as a surviving strategy for pre-existing conditions [34]. Most of the epidemiologic studies explored the association between ELF EMFs and the susceptibility to different cancers. In the next section the residential/domestic and the occupational exposure to MF as related to cancer occurrence will be described. 

### 2.1. Epidemiology of Residential/Domestic Exposure to MF

A survey of the literature indicates that residential exposure to EMFs is associated to an increased risk of cancers, particularly breast cancer, brain tumors, and leukemias. However, most of these studies are based on small numbers of high field-exposed cases [16,35,36] and an increasing number of studies does not support an epidemiologic association of adult cancers with residential MFs [35,36,37]. Many of the studies clearly have shortcomings, which often prevent any firm conclusions [38,39]. Moreover, the indirect measures of EMF exposure used may also correlate with other factors such as social status (e.g., age, race, gender) or environmental pollution. It is possible that these unconsidered and confounding factors may contribute to the cancer rates that are reported and also to the contrasting results that are reported in various EMF studies [40]. Indeed, EMF, which itself is not believed to be genotoxic, could influence carcinogenesis if it exerted either direct or indirect effects on target cells [41]. 

Another important issue is the exposure assessment. The exposure to MF can vary greatly over time and distance, has multiple sources, and is imperceptible and ubiquitous [42,43,44,45]. In exposed schools, children may experience a higher chance of receiving a mean exposure >0.4 µT during school hours [46,47]; whereas those living in big buildings or using electric heating appliances in larger families had a generally higher level of personal indoor exposure [48]. Based on the known location of domestic and service MF sources, apartments can be reliably classified as high and low MF-exposed [49,50,51]. Methodologies for estimating MF at study residences as well as for characterizing the sources of uncertainty in these estimates have been developed [52]. In residential/domestic epidemiological studies, geographic information that is collected in an exact place of residence at the time of cancer diagnosis can provide several strategic geophysical elements for assessment [53]. The estimation of the overall exposure level from a single address is also informative [54]. In general, the public health impact of residential fields is considered limited, but the available data show the occurrence of both no impact and substantial impact possibilities [55].

#### 2.1.1. Brain Tumor

The incidence of primary brain tumors has increased in many countries worldwide [56,57] and gliomas are the most frequent primary brain tumors in adults [58]. Residential exposure during childhood to EMF produced inconsistent results and a lack of an association when related to brain cancers, regardless of the exposure metrics that were used whether based on wire codes, distance, or the measured or calculated fields [59]. When focusing on the health effects, the most studied sources of ELF MF are power lines. Exposure to ELF MF that is emitted by power lines can be assessed by direct methods that rely on measurements at a given place over a time range [60] or by individual monitoring through measuring ELF MF exposures throughout the day by wearable dosimeters [61]. Both methods give little or no information on historical exposure to ELF MF. Indirect methods include geographical information system (GIS) which have been used along with declarative data, such as residential history, to assess residential ELF MF exposure in the general population [62,63,64,65]. Case-control studies that are based on death certificates revealed an association between adult brain tumor mortality and living less than 50 m (odd ratios, OR 1.10 95%, CI 0.74–1.64) [66] or 100 m (OR 2.99, 95%, CI 0.86–10.40) [67] from power lines. In a recent work, significant associations were found between the cumulated duration living at <50 m to high voltage lines (50/60 Hz, 0.3 μT) and all brain tumors (OR 2.94; 95%CI 1.28–6.75) and glioma (OR 4.96; 95%CI 1.56–15.77) [65], confirming previous studies [35,66,67,68,69]. Contrasting results have been reported for brain tumors in children. In childhood brain cancer, with the exception of the possibility of a moderate risk increase in high cut-point analyses (0.3/0.4 µT), no increased risk was evident for different exposure metrics [70,71], whereas children whose MF exposure level was above 0.3 or 0.4 μT, an elevated risk of brain tumor was observed [72,73,74]. In residential areas, the transient electric and MFs would induce higher current density in the child‘s body than power frequency fields with similar field strength [75]. In some studies, there is no evidence for a role of ELF cellphone EMFs in childhood brain cancer [27]. 

#### 2.1.2. Breast Cancer

Breast cancer threatens women with the highest incidence and second highest mortality rate of all cancers and in women aged 65 and older when nearly one half of all new breast cancer is diagnosed [76]. The excessive exposure to MFs increases the risk of female breast cancer, as demonstrated in several pooled or meta-analyses as well as subsequent peer-reviewed studies [69]. It is questionable whether chronic human exposure to MFs might affect melatonin secretion, its circadian rhythm, or both [77,78,79,80]. In general, no cumulative effects on melatonin secretion in humans have been found in response to MFs and this rebuts the “melatonin hypothesis” in which a decrease in plasma melatonin concentration (or a disruption in its secretion) would be correlated with the occurrence of breast cancers as a consequence of exposure to MFs [81,82,83,84,85,86,87,88]. Indeed, MF exposure correlates with an increased proliferative activity of the mammary epithelium, which is a likely explanation for the cocarcinogenic or tumor-promoting effects of MF exposure that is observed [89]. However, some authors found a motivation for going back to the melatonin hypothesis in relation to data, suggesting magnetosensory disruption by ELF MF in mammals, and magnetosensitivity in humans, along with the influence of MFs on circadian rhythmicity with a consequent disruption of non-photic sensory stimuli of various nature [90].

In studies that were based on the measurement of an electric bedding device, only premenopausal breast cancer (OR = 1.23; 95% Cl: 1.01, 1.49, *p* = 0.04) showed a slight increased risk [91]. Breast cancer was found to increase with the number of years and seasons of bedding device use during sleep. Similar trends in dose response were shown in both premenopausal and postmenopausal women and for both estrogen receptor-positive and estrogen receptor-negative tumors. Therefore, there is a growing body of evidence that the use of electric bedding devices may increase breast cancer risk [92]. In terms of geographic variation of breast cancer rates, the results are inconclusive and do not support a major role of MF risk factors in the etiology of breast cancer [93].

#### 2.1.3. Leukemias

Leukemia is the most common cancer in children [94]. The analysis of reports on childhood leukemia as related to exposure to MFs shows that a statistically significant association between MF exposure and childhood leukemia is found in almost all government or independent studies with an elevated risk of at least OR = 1.5, while a not significant or even suggestive association is reported in many industry supported studies [69].

Several meta-analyses showed a statistical association between childhood leukemia and a range of exposure 0.1–2.36 µT MF intensity [74,95,96,97,98,99]. Children that are exposed to elevated ELF-MF show relative risks of leukemia between 1.3 and 2 [100,101,102] and the highest exposure was associated with an increased risk of B-lineage acute lymphoblastic leukemia (B-ALL) when compared with lower exposures [103]. A significant association was observed during the night (OR = 3.21, 95% CI 1.33–7.80) between childhood leukemia and MF exposure [104].

Several epidemiologic case-control studies examined the association between childhood cancer risk and distance to high-voltage overhead transmission lines (HVOTL). Statewide, record-based case-control studies of childhood leukemia evidenced the occurrence of risk that was associated with greater exposure to MF that was generated in areas that were close to power lines [66,105,106,107,108]. Living in polluted regions and pre- and post-natal exposure to high voltage power lines has been described as risk factors of acute lymphoblastic leukemia (ALL) in people of low socioeconomic status Iranian population [109]. In children with ALL, ELF MF-exposure was found to have no impact on the survival probability or risk of relapse [110].

The occurrence of childhood acute leukemia (AL) has been studied around nuclear power plants (NPP). The results suggest a possible excess risk of AL in the close vicinity to NPP [111]. The small, but statistically significant increased incidence of AL in the surrounding of some NPP have motivated governments to work toward a better understanding of the main causes of AL long-term strategic research agendas through interdisciplinary and international efforts [112].

MFs are not the only factor that varies in the vicinity of MF sources, complicating interpretation of any associations. Several reports demonstrate that MFs that are generated by different sources are not an important cause of leukemia both in adults [37,113,114] and children in many geographical areas [12,113,114,115,116,117,118,119,120]. Moreover, exposure levels in some big cities are always significantly far lower than 0.3–0.4 µT [121].

The associations that were observed between power lines exposure and childhood leukemia appears to be not related to mobility [122,123]. No indications were also found of an association of risk for people that are exposed to magnetic fields from underground [124] and above ground lines [106,107] or of a trend in risk with increasing MF for leukemia. Residential proximity to transformer stations has been associated with a borderline risk of childhood cancer [125]. 

Distance from HVOTLs during the year of birth is unlikely to be associated with an increased risk of leukemia [73] and little evidence was found between exposure to MF inside infant incubators and increased risk of childhood leukemia [126]. No statistically significant association was observed between wire codes and childhood leukemia [127]. Recently, a synoptic analysis provided evidence that the risk of childhood leukemia is not increased by exposure to ELF EMFs, suggesting that IARC’s classification of ELF EMF needs revision [128].

By considering both significant and not significant correlations between MF and leukemia, no major environmental risk factors (including MFs) have been established as major contributors to the global leukemia burden, although distinct incidence patterns by geography, age, and sex suggest a role of the environment in its etiology [129]. The results may be affected by several sources of bias: analyses that are based on continuous exposure show no exposure-disease association, while incoherent exposure-outcome relationships characterize analyses that are based on categorical variables [130]. 

#### 2.1.4. Other Cancers

EMF in the home-environment (color tv, computer monitor, microwave-oven, cellular phone, etc.) might act as potential contributing factors for the development of cancer, as well as exert indirect effects on humans. Microwave exposure induces L-amino acids change to D-amino acids, and exposure of the human body to microwaves over a long period of time may contribute to induction of cancer [131]. Exposure to EMF that is generated by electric blankets has been suggested to increase the risk of hormone-dependent cancers such as testicular [132] and prostatic [133], but was not significantly associated with endometrial cancer risk [134]. MFs of industrial frequency have been shown to be a risk factor for occurrence of oncological diseases in the population and an increased incidence of malignant tumors has been noted as the induction of the magnetic field that is produced by HVOTL increases [135]. The incidence of melanoma has been linked to the distance to FM broadcasting towers. A correlation between melanoma incidence and the number of locally receivable FM transmitters was found when geographic differences in melanoma incidences were compared with the magnitude of this environmental stress. Therefore, melanoma might be associated with exposure to FM broadcasting [136]. By considering pregnancy duration, neonatal birth weight, length, head circumference, gender, and con-genital malformations, no significant effects of ELF EMFs were observed; however, precautionary measures are necessary to reduce exposure to EMFs by pregnant women [137].

The Appendix A reports the relationship between MFs and cancer in epidemiologic studies that are related to domestic/residential exposure to MFs. The type of cancer, study design, source of MF, range of MF exposure, location of the epidemiologic study, and the main conclusions are reported. Figure 2 summarizes the epidemiology of residential/domestic exposure to MF.

### 2.2. Epidemiology of Occupational Exposure to MF

During the last few decades, the intensity level of the EM occupational environment has dramatically increased. By job category, the most highly exposed jobs (>0.23 µT) included electronics workers, cooks and kitchen workers, cashiers, bakery workers, textile machine operators, and residential and industrial sewing machine operators [138]. The main components of EM pollution are in the ELF (10–300 Hz) and in ultra-low (ULF: 0–10 Hz) frequency bands [139]. Occupational epidemiology reveals that exposure to ELF EMF is generally greater than that in the general population and concerns a large number of workers in a variety of industries (see [140,141] for a historical overview of the occupational EMF epidemiology).

MF exposure limits are more than a thousand times higher than the magnitudes that are associated with the cancer risks that are observed in epidemiologic studies, leaving millions of workers exposed to MF in this large gray area where the public health consequences are unclear [140,141]. The International Labor Organization defines occupational exposure as “exposure of a worker received or committed during a period of work” [142] while the ICNIRP defines occupational exposure as “all exposure to EMF experienced by individuals as a result of performing their regular or assigned job activities” [143].

As discussed for residential/domestic exposure to EMF, in the case of occupational exposure, contrasting results have also been presented in relation to the co-occurrence of different cancers including brain and breast cancer and hematological malignancies.

#### 2.2.1. Brain Cancer

Occupational exposure to ELF EMF is a suspected risk factor for brain tumors; however, the literature reports contrasting results. In adults, some meta-analyses of occupational studies indicate a slightly higher risk for electrical workers, suggesting a small increase in brain cancer risk [61,144,145], including childhood brain tumors [146], while others found no evidence to support the hypothesis that exposure to MFs is a risk factor for gliomas or meningiomas [147,148,149,150].

#### 2.2.2. Breast Cancer

Several studies have evaluated the evidence linking women’s occupation and workplace exposures to breast cancer. Overall, the data do not suggest that occupational exposures to EMF increases the risk of breast cancer [144,145,151,152] with some exceptions [153]. Some studies have found an effect when looking at overall risk elevations in the women studied [154] with increased risk among postmenopausal women but not premenopausal women [155]. The hypothesis that daytime occupational exposure to MF enhances the effects of nighttime light exposure on melatonin production (see above the “melatonin hypothesis”) has been provided [153]. Occupational MF exposure has also been suggested as a risk factor for breast cancer in men. An elevated risk of breast cancer was found in men who were exposed to 0.6 T when compared to those with exposures <0.3 T; however, large case-control studies of breast cancer in men that have been conducted to date provide limited support for the hypothesis that exposure to MF increases the risk breast cancer in men [156].

#### 2.2.3. Leukemias

Epidemiological studies addressing occupational ELF MF exposure and risk of leukemia in adults have yielded slightly increased risks in exposed workers. Some studies show a positive correlation between occupational MF and leukemias [157,158,159,160] and have suggested that stronger effects may be observed for acute myeloid leukemia (AML) [161,162], chronic myeloid leukemia (CML), ALL [138], lymphatic leukemia (LL) [163], and for chronic lymphocytic leukemia (CLL) [162]. In general, however, no clear exposure-response pattern has emerged from the studies that evaluated exposure levels and some results do not support an association between occupational ELF MF or electric shock exposure and AML [164].

Differences between the study designs or the populations that were studied might be the cause of lack of consistency regarding the type of leukemia that is associated with MF exposure, and still no firm conclusions can generally be drawn based on the existing evidence [165,166,167]. No association was found between childhood leukemia risk and parental occupational exposure to ELF EMF [168,169,170].

#### 2.2.4. Other Cancers

In men, exposure to EMF showed an increased incidence of tumors of the liver, biliary passages, kidney, and pituitary gland; for these cancer sites an exposure-response relation was indicated [171]. There was very limited evidence for associations between occupational ionizing radiation and testicular cancer, while there were some positive associations for EMF [172].

Women that were exposed to MF showed an increased incidence of astrocytoma I-IV, for cancer of the corpus uteri and multiple myeloma, with a clear exposure-response pattern [171]. 

For both men and women, there was weak support for the hypothesis that occupational MFs exposure increased the risk of non-Hodgkin lymphoma [173], acoustic neuroma [174,175], and thyroid cancer [176], while sources that produce ELF fields were not associated with neuroblastoma in offspring [177].

MF has previously been associated with mortality from acute myocardial infarction (AMI) and arrhythmia but not from chronic coronary heart disease (CCHD) or atherosclerosis in electric utility workers. For cumulative exposure, no association was observed with mortality from AMI or CCHD, indicating no exposure-related risk increase for AMI mortality, which does not confirm previous results [178].

Appendix A reports the relationship between MFs and cancer in epidemiologic studies that are related to the occupational exposure to MFs. The type of cancer, study design, source of MF and occupation, range of MF exposure, location of the epidemiological study, and the main conclusions are reported. Figure 3 summarizes the epidemiology of occupational exposure to MF. 

## 3. In Vivo and In Vitro Effects of Magnetic Fields on Cancer

In many studies ELF exposure causes significant changes in cell survival, cell cycle progression, DNA integrity, and proliferation [40,179]. The cellular response to ELF MF may depend on many parameters including osmolarity, frequency, waveform, the strength and the exposure duration of the electromagnetic field, genetic/biological characteristics of the cells [180], specific metabolic state, or the specific stage in the cell cycle [180,181]. On the other hand, a common effect of exposure to SMF is the promotion of apoptosis and mitosis, but not of necrosis or modifications of the cell shape [182,183]. The unbalance of the apoptotic process could be linked to Ca^2+^ fluxes that are, in turn, dependent on the effect on the plasma membrane that is exerted by SMF [184,185,186,187]. Other possible effects of SMFs that may lead to perturbation of the apoptotic rate, such as an alteration of the gene pattern expression or the increase of oxygen free-radicals, could be, in turn, a co-carcinogenic factor leading normal cells, most likely with other sub-lethal changes, to contribute to the development of diseases [183]. 

The reduction of cell proliferation due to MF has been attributed to the interference of signal transduction processes due to the tangential currents that are induced around the cells [188]. The poor reproducibility might be caused by the period-dependent converse cell growth due to the MF and might explain the adverse effects that are observed in several experimental investigations [189]. 

Quantitative analyses of protein kinases C (PKC) expression patterns demonstrated the translocation of PKC from the cytosolic to the membrane fraction was not affected by MFs [190]. The phosphorylation of extracellular signal-regulated kinases 1/2 (ERK1/2) is increased in response to ELF MF; however, the small increase in ERK1/2 phosphorylation is probably insufficient to affect proliferation and oncogenic transformation [191]. Furthermore, repeated ELF EMF exposures did not show consistent response profiles to time courses of immediate early genes, apoptotic genes, cell proliferation, and stress response [192].

Small changes in transcription may occur in response to MFs [193,194]. Exposure to 900 MHz identified a differential expression of functional pathways genes [195]. Indeed, epigenetic changes, including modifications of histones and microRNA expression and DNA methylation, can be associated to ELF MF exposure [196,197,198]. However, in HeLa cells, RNA polymerase-catalyzed RNA synthesis as well as DNA polymerase-catalyzed DNA synthesis were found to not be statistically significantly affected by 60 Hz 0.25–0.5 T exposure for 0–60 min [199]. 

Aberrantly-expressed serum exosomal miRNAs upon MF exposure suggests a series of informative markers to identify the exact dose of MF exposure [200]. ELF MF exposure stabilizes active chromatin, particularly during the transition from a repressive to an active state during cell differentiation [201]. Membrane receptors could be one of the most important targets where ELF MF interacts with the cells [202] and RAS proteins, a member of a large family of GTP-binding proteins that are involved in intracellular signal pathways, may participate in the signal transduction process of 50 Hz MF [203]. EMF was also found to affect the proteasome functionality, inducing an increase in its proteolytic activity [204].

To answer how MF may cause cancer, the action of MF as mutagenic agents and MF involvement on chemical reactions that generate free-radicals have been considered. MF does not seem to exert mutagenic effects and the generation of free-radicals that might be linked to several other factors, beside the variability of MF exposure [40]. In the next part of this section, the current data that are related to studies on human in vitro and cell-free systems as well as in animal models will be discussed. This section will also discuss the role of reactive oxygen species (ROS) and reactive nitrogen species (RNS) and the role of radical pair (RP) formation in MF-cancer relationship.

### 3.1. Studies on Human Cells (In Vitro Cellular Studies and Cell-Free Systems)

Exposure to ELF MFs combined with ionizing radiation do not suggest any synergistic or antagonistic effects on human blood cells [205], whereas, by using sister chromatid exchanges and comet assays, a slight but significant decrease of cell proliferation was evident in blood cells that were exposed to 980–1020 µT [206].

Dermal fibroblasts that were exposed to 1 mT from one to five hours showed a reduction of their viability [207], while the rePETitive exposure to MF with ELF induced DNA double-strand breaks and apoptosis in lung fibroblasts [208]. However, a longer exposure (48–72 h) to a reduced MF intensity (20–500 µT) did not result in any appreciable effect in the structural morphology and proliferation of human fibroblasts [209].

Exposure of glioma cells to MF of 1.2 µT intensity for three hours revealed changes in both gene and protein expression. Microarray results showed an up-regulation of five genes, whereas 25 genes were down-regulated upon MF exposure, suggesting a response in the glioma cells to the MF treatment [210]. Proteomic studies on glioma SF767 cells show a cytoskeletal intermediate filament protein increased following a low-level MF [211]. On the other hand no mechanisms that would explain the reported association between MF and carcinogenesis were observed in H4 glioma cells [212].

In human keratinocytes, ELF EMF induces a slight oxidative stress that does not overwhelm the metabolic capacity of the cells or have a cytotoxic effect when the cells are exposed to 25–200 µT MF intensity [213] but it does not affect melanin synthesis or skin pigmentation [214].

The adherence of leucocytes and T-lymphocytes that were taken from cancer patients strongly increased following exposure to sinusoidal MF at 50 Hz and 0.5–10 mT MF intensity for one hour [215,216], whereas a low-frequency pulsing electromagnetic field of 45 mT for three hours induces cell death in native peripheral blood mononuclear proliferating cells that were isolated from AML patients [217].

ELF MF increases the rate of cell death in normal human lymphoblastoid cell lines and is ineffective on genetic instability syndromes cell lines, i.e., Fanconi anemia group C and ataxia telangectasia, suggesting that the response of cells to ELF MF is modulated by genes that are implicated in genetic instability syndromes [218]. Exposure to a pulsating MF 50 Hz, 45 mT three times for three hours was found to protect U937 human lymphoid cell lines against puromycin-induced cell death in a cell density-dependent manner (an increased density induced cells death and prevented puromycin-induced cell death) [219].

The effects of 6- and 10-T SMFs was studied on HL-60 cells to evaluate the expression of protooncogenes. It was found that exposure to a strong MF gradient induced c-Jun expression, whereas no alteration of the expression of the *c-fos* and *c-myc* protooncogenes was observed [220]. In the same cell line, exposure to LF EMF to lower intensity (5, 300, and 500 µT) did not affect calcium signaling [221].

An ELF magnetic field was also found to influence the early development of mesenchymal stem cells that were exposed for 23 days to 50 Hz and 20 mT intensity [222], whereas no variations in surface morphology and cell death occurred between the control and the exposed osteosarcoma human cells that were exposed to 50 Hz and 0.5 mT, although significant changes were noted in cell growth [223].

Appendix A reports the effects of MFs on human cell lines. The type of cell, response to MF, range and duration of MF exposure, materials and methods that were used, and the main conclusions are reported.

### 3.2. Studies on Animals

Exposure to MF was found to be a cancer promoting factor in animal experiments; animal studies are often used in the evaluation of suspected human carcinogens [224]. However, discrepancies in results were found to be associated with the use of different sub-strains, different sources for diet, environmental conditions, and MF exposure metrics [225,226,227]. The continuous monitoring of both MF and other environmental parameters is, therefore, an important part in the overall quality of the obtained results [228]. Experiments with animals were also important to determine which genetic and environmental factors are critical for potential carcinogenic or cocarcinogenic effects of ELF EMF exposure [229]. Furthermore, the genetic background was found to play a pivotal role in effects of MF exposure [230]. To study the effects of MFs on cancer cell growth, multiple exposure levels have been performed using mice and rats as experimental models. 

#### 3.2.1. Studies on Mice

Several studies revealed that there were not significant effects of MF on cancer in mouse studies, failing to support the hypothesis that acute MF exposure causes DNA damage. The experimental mice were injected with mammary murine adenocarcinoma to investigate the interaction between a 50 Hz, 2 mT MF exposure and cell growth. 

Neither the time of tumor cell injection nor the time of exposure produced differences between the unexposed, sham, and exposed mice [231] and no association between exposure and the incidence of benign or malignant tumors was found in squamous cell carcinomas of mice that were exposed to 60 Hz, 2 mT [232]. Also, the results do not support the hypothesis that acute MF exposure causes DNA damage and apoptosis in the cerebellums of immature mice that were exposed to 60 Hz and 1 mT for 2 h [233,234].

Exposure to 50 Hz and 500 µT MF did not alter responses of inflammatory genes and the activation of splenic lymphocytes in mice [235] and was not a significant risk factor for hematopoietic diseases, even when high exposure levels were used (50 Hz, 1 mT for 7 days) [236]. In mice, as revealed in humans (see above), long-term and continuous exposure to simulated powerline MFs (0.5–77 µT) did not result in a decreased nocturnal melatonin secretion [85].

The use of NI MFs has shown early promise in a number of animal studies as an effective tool against many types of cancer (see also below). The effect of varying durations of MF exposure on tumor growth and viability has been studied in mice that were injected with breast cancer cells by using an in vivo imaging system. The results showed the potential of MF in cancer therapeutics, either as adjunct or primary therapy [237].

Long-term exposure showed a significant effect of MF on mice. When a parental generation of six week-old OF1 mice was exposed to an artificial ELF MF (50 Hz, 15 µT) for 14 weeks, activated partial thromboplastin time and reptilase time values significantly increased in the female mice, which also showed a very significant shortening of the prothrombin time that was associated with ELF MF exposure [238]. A chronic exposure of mice to a 60 Hz and 110 µT intensity was found to influence some hematologic parameters and the weight of thee liver and also caused spleen hyperfunction [239]. Long-term exposure to MF (50 Hz, 50 µT) was found to be a significant risk factor for neoplastic development and fertility in C57BL/6NJ female mice and C3H/HeNJ male mice [240].

#### 3.2.2. Studies on Rat

Both positive and negative effects of MF have been demonstrated in studies using rats. Artificial MFs of 1 mT that was administered for seven weeks was not carcinogenic nor cancer-promoting for colon carcinogenesis in male Sprague–Dawley rats [241]. No overall effects of 60 Hz and 1 mT MF on splenomegaly or survival were found in the exposed Fischer/344 rats. In addition, no significant and/or consistent differences were detected in the hematological parameters between the exposed and control rats [242]. MF exposure for 14 weeks to 6.45 µT did not appear to be a strong co-tumorigenic agent in Sprague–Dawley female rats mammary, lung, and skin models [243]. On the other hand, EMFs resulted in significant alterations in cell adhesion mechanisms when histological, immunohistochemical, and histopathological analyses were performed on Wistar albino rats that were exposed for six months to 5 mT MFs [244]. MF exposure (50 Hz, 100 µT for 2–26 weeks) of female Sprague–Dawley rats resulted in an increased proliferative activity of the mammary epithelium which was associated with the cocarcinogenic or tumor-promoting effects of MF exposure that was observed in the 7,12-dimethylbenz(a)anthracene model of breast cancer [89], and mammary tumorigenesis [245].

The Appendix A reports the effects of MFs on both mice and rats. The type of cell, response to MF, range and duration of MF exposure, materials and methods used, and the main conclusions are reported. Figure 4 summarizes the in vivo and in vitro effects of magnetic fields on cancer from studies on human cells (including cell-free systems) and in animals (mice and rat).

### 3.3. Involvement of Reactive Oxygen Species (ROS) and Reactive Nitrogen Species (RNS) 

ROS (such as superoxide [O_2_^•^^−^] and hydrogen peroxide [H_2_O_2_]) and RNS (e.g., nitric oxide [NO^•^]) are generated during the oxidative cell metabolism. The cellular oxidative stress depends on the balance between the production of ROS and RNS and the activity of the antioxidant system. Excessive ROS/RNS, which is caused by the deregulated redox homeostasis, is a hallmark of disease [246]. Free-radical-scavenging enzymes, such as catalase (CAT), glutathione peroxidase (GPX), and superoxide dismutase (SOD), are the first line of defense against oxidative injury [247]. After EMF exposure, significant variations in the total antioxidant activity; vitamin E and vitamin A concentrations; increase of malondialdehyde (MDA, a product of polyunsaturated fatty acid peroxidation which is used as an indicator of oxidative stress in cells and tissues); and plasma selenium concentration in erythrocytes have been observed [248]. A recent review concluded that most animal and many cell studies show increased oxidative stress that is caused by MF [249]. Free-radical formation and the consequences of their effects on living systems explains the increased cancer risks that are associated with mobile phone use, occupational exposure to MF, and residential exposure to power lines [250].

ROS may be involved in RP reactions that have been considered as one of the mechanisms of transduction that is able to initiate EMF-induced oxidative stress [19]. It is known that applied MFs and magnetic isotope substitution can alter the rates and product yields of free-radical reactions with the formation of transient paramagnetic intermediates in non-equilibrium electron spin states. The most common source of spin-chemical effects are organic RPs. Typically formed in a singlet (S) or a triplet (T) state by a reaction that conserves electron spin, RPs interconvert coherently between their S and T states as a result of the Zeeman, hyperfine, exchange, and dipolar interactions of the electrons and the nuclear spins to which they are coupled [251]. Applied MFs alter the extent and timing of the S⇆T interchange and hence the yields of products that are formed spin-selectively from the S and T states [252]. MFs and spin effects have proven to be useful mechanistic tools for radical mechanism in biology [252] and the RP mechanism (RPM) has been associated with increased levels of ROS [253]. Moreover, the role of cryptochromes as the putative magnetosensitive molecules in magnetoreception has been considered in the RPM and discussed as related to cancer-relevant biological processes [254]. The results are also consistent with MF effects on light-independent radical reactions [255].

Significant increases in ROS levels have been found to influence the hepatic redox state [256] and were observed in several cell lines just after the end of ELF EMF exposure [213,257,258,259,260,261,262,263,264,265]. ELF EMF exposure also elevates the expression of RNS and O_2_^•−^, which are countered by compensatory changes in antioxidant CAT activity and enzymatic kinetic parameters that are related to cytochrome P450 (CYP-450) and CAT activity [266]. Moreover, the modulation in kinetic parameters of CAT, CYP-450, SOD, and MDA concentration and iNOS enzymes in response to ELF EMF [267,268,269] and the negligible effects on GPX [270] indicates an interaction between the ELF EMF and the enzymological system. SMF promoted the release of ferrous iron (Fe^2+^) and induced the production of ROS in osteosarcoma stem cells [271]. Superoxide increased in the micronucleus and mitochondria with an exposure-response relationship and cytosolic superoxide increased at 10 µT in SH-SY5Y and C6 cells [272]. These results confirm that the threshold for biological effects of ELF MFs may be as low as 10 µT.

In liver tissue of female rats, long-term ELF-MF exposure enhanced the oxidative/nitrosative stress and might have a deteriorative effect on cellular proteins rather than lipids by enhancing 3-nitrotyrosine formation [273]. MF appears to induce apoptosis effects through oxidative stress [274] and mitochondrial depolarization [275], whereas the influence of correlations between ELF EMF and vitamin E supplementation have been shown on antioxidant enzyme activity in AT478 malignant cells in vitro [276]. In embryonic stem cell-derived embryoid bodies, exposure to 10 mT MFs increased ROS [277]. 

Appendix A reports the effects of MFs on ROS and RNS. The type of cell/tissue/organ, response to MF, range and duration of MF exposure, materials and methods used, and the main conclusions are reported. Figure 5 summarizes the involvement of ROS and RNS in the cellular and organismic responses to MFs.

## 4. Magnetic Fields and Cancer Theranostics

So far, we have discussed the possibility that exposure to MFs may be correlated with cancer. However, MFs have been widely used for cancer diagnostic and therapeutic applications. In this section, the use of MRI as a cancer diagnostic and therapeutic method and the use of magnetic nanoparticle for cancer treatment will be discussed.

### 4.1. Magnetic Fields and Cancer Diagnosis

MRI, a medical application of nuclear magnetic resonance (NMR), uses strong MFs, MF gradients, and radio waves to generate images of the organs in the body [278]. There is a trade-off between MF dose effects and the image quality of MR-guided radiotherapy systems and the MF strength may affect the severity of MF dose effects [279], such as the electron return effect [280]. MRI has also been used in combination with positron emission tomography (PET) and has a strong potential clinical use for the imaging of several forms of cancer [281]. Using ultrasmall superparamagnetic particles of iron oxide (USPIO), it is possible to enhance the power of MRI for noninvasive diagnostics of different types of cancer [282,283] (see also 4.2). Since alterations in the Na^+^ ion concentration may potentially be used as a biomarker for malignant tumor diagnosis and prognosis, Na-23-magnetic resonance imaging (Na-23-MRI) was found to have potential as a direct noninvasive in vivo diagnostic and prognostic biomarker for cancer therapy, particularly in cancer immunotherapy [284]. The potential role of MRI in the detection of several cancers will be discussed in the next paragraphs.

#### 4.1.1. Brain and Glioma Cancer

MRI has allowed the characterization and diagnosis of human brain cancers in spatial and volumetric analysis [285], as a substitute for biopsies [286], in glioma genotyping [287], brain cancer classification [288,289], as a non-invasive tool for simultaneous and automated tumor segmentation [290], and to investigate the early stages of slow-growing invasive tumors [291,292,293,294]. MRI is used before treatment and at the end of treatment or disease progression [295,296] and to assess neurological complications of cancer treatment [297]. PET/MRI is used in brain tumor grading and staging [298,299] as a diagnostic and therapeutic strategy for glioma [300] and for improved diagnostic and therapeutic assessment in pediatric, teenage, and young adult brain tumors [301]. ^11^C–methyl-L-methionine (C-11-MET) PET/MRI was found to improve the diagnostic accuracy to differentiate treatment-related changes from true progression in recurrent glioma [302] and was useful for the assessment of isocitrate dehydrogenase (IDH) status [303]. Perfusion MRI is used to differentiate glioma from brain metastasis [304], whereas dynamic glucose-enhanced (DGE) MRI is a feasible technique for studying brain tumor enhancement reflecting differences in tumor blood volume and permeability with respect to a normal brain [305]. MRI-coupled fluorescence molecular tomography (MRI-FMT) determines epidermal growth factor receptor status in brain cancer [306] while advanced MRI techniques contribute to biological and imaging features of glioma and immune system interactions [307] and in the clinical management of adult gliomas [308]. The application of gadofluorine-M (GfM) results in superior delineation of experimental glioma compared with conventional MRI techniques [309], whereas the labeling tumor cells with superparamagnetic iron oxide (SPIO) and performing an MRI scan dynamically monitors the development and biological behavior of glioma at a very early stage [310]. Vascular, extracellular, and restricted diffusion for cytometry in tumors MRI has been used as a potential test for diagnostic stratification to investigate the tissue microstructure in glioma [311], whereas dynamic contrast-enhanced (DCE) MRI detects increases in gadolinium (Gd)-enhancement of brain tumors [312] and provides an unambiguous indication of the brain tumor photodynamic therapy outcome [313]. The enhancement on MRI may assist in identifying HER2 overexpression in breast cancer brain metastases [314].

#### 4.1.2. Head and Neck Cancer

According to the National Cancer Data Base, head and neck cancer accounted for 6.6% of all new cancers [315]. The use of PET and MRI, separately or combined, has been successful for assessing the metastatic lymph nodes in patients with head and neck cancer and offers advantages in staging with regard to increased anatomical details and radiation dose reduction [316,317,318]. Diffusion-weighted imaging (DWI) and intravenous (IV) contrast T1 dynamic perfusion imaging are a valid support for the functional MRI of tumors of the head and neck [319,320,321,322] and algorithms for automatic head and neck 3D tumor segmentation from MRI have been developed [323]. For head and neck cancer, MRI-guided radiotherapy achieves clinical outcomes that are comparable to contemporary series reporting on intensity-modulated radiotherapy (IMRT) [324] and the use of targeted 3 T MRI was found to be useful for defining the presence and extent of large nerve perineural spread in head and neck cancers [325]. Retrospective image fusion of PET/MRI for the assessment of the extent of the primary tumor (T stage) and metastasis to regional lymph nodes (N stage) has been evaluated [326], whereas USPIO-enhanced MRI in patients with a clinical neck cancer was able to differentiate borderline-sized lymph nodes [327].

#### 4.1.3. Thyroid Cancer

Thyroid cancer is one of the fastest growing cancer diagnoses worldwide [328]; it is the most common endocrine malignancy in children [329], it is three times more frequent in females [330], and poorly differentiated and anaplastic thyroid carcinomas represent a challenge to physicians on the basis of the current cancer treatment modalities [331]. MRI shows sensitivity and specificity to diagnose recurrent thyroid carcinomas [332], thyroid cancer cervical lymph node metastases [333], papillary carcinomas [334], and is more sensitive than ultrasonography in detecting central compartment metastases in papillary thyroid carcinoma [335]. However, MRI is inadequate for the detection of metastases in small lymph nodes [336]. Esophageal invasion by thyroid carcinoma was accurately predicted with MRI [337], whereas MRI, DWI-MRI, and MRI-based computer-aided diagnosis (CAD) allow the differentiation of thyroid nodules whether benign or malignant [338,339,340] and for the detection of inner and outer thyroid lamina invasion [341]. PET/MRI of the neck is superior to PET-computed tomography (CT) in detecting iodine-positive lesions [342] and provided further information in an overwhelming majority of thyroid cancer patients [343]

#### 4.1.4. Breast Cancer

Among current clinical imaging modalities, MRI of the breast has the highest sensitivity for breast cancer detection and is becoming an indispensable tool for breast-imaging procedures [344,345]. Multi-parametric (Mp) MRI is the most sensitive imaging modality for breast cancer detection [346] and has been successfully used in combination with PET/MRI [347] and other imaging modalities. Abbreviated breast MRI uses only a select number of sequences and postcontrast imaging reduces the table time and reading time to maximize availability, patient tolerance, and accessibility [348], and may enable the more widespread use of breast-MRI for screening [349].

#### 4.1.5. Lung Cancer

Among all cancer, lung cancer has the highest rate of mortality in the western world [350]. MRI is suitable for lung cancer screening with an excellent sensitivity and specificity of malignant as well as calcified and subsolid nodules [351,352,353,354,355] and for radiation treatment planning in lung cancer [356]. DWI-MRI protocol have been designed for imaging malignant lung tumors, achieving satisfactory within-patient repeatability [357], while recent advances in PET/MRI for lung cancer staging have been reviewed [358,359].

#### 4.1.6. Gastric Cancer

Gastric cancer is an important healthcare problem from a global perspective, being the fifth most common malignancy and the third leading cause of cancer-related death [360]. Most of the cases are diagnosed at late stages when the treatment is largely ineffective, and MRI is of great value in patients with gastric cancer [361,362]. MRI is useful for distant metastasis assessment with particular reference to peritoneal and liver metastases assessment [361]. DWI-MRI and apparent diffusion coefficient (ADC) values showed to be useful in preoperative MRI staging of gastric cancer [363,364], but has a low accuracy to detect or to differentiate metastatic and non-metastatic lymph nodes based on their ADC values in gastric cancer [365]. MRI is useful for the diagnosis of serosal invasion of gastric cancer [366], in the diagnosis of regional lymph node metastases [367], and is the best method in the assessment of gastric wall infiltration in gastric cancer [368]. MRI is more accurate in achieving adequate staging results [369,370] and in the evaluation of T-stage than CT [371,372]. DCE-MRI parameters of gastric cancers may provide prognostic information [373,374], whereas multiparametric fully integrated ^18^fluordesoxyglucose ([F-18]-FDG)-PET/MRI may improve the diagnostic accuracy for translational gastric cancer research [375], for preoperative M staging as well as the resectability of gastric cancers compared to multi detector computed tomography (MDCT) [376]. Compared with PET/CT, PET/MRI performs more accurately in TNM staging and is optimal for accurate N staging [377], whereas high-resolution MRI (HR MRI) has good diagnostic performance and may serve as an alternative technique in the T staging of patients with esophagogastric junction cancer in clinical practice [378]. The preoperative prediction results of MRI are consistent with postoperative pathological results [379], although the clinical use of MRI for gastric cancer is still under discussion [380].

#### 4.1.7. Pancreatic Cancer

Pancreatic cancer is a deadly disease, mainly because it is very resistant to chemotherapy and radiation therapy and is generally discovered very late [381]. MRI provides relevant information for the diagnostic evaluation of malignant pancreatic tumors [382,383] by predicting the survival in advanced pancreatic cancer patients [384]. The use of MRI of the liver for the initial staging of pancreatic cancer results in lower total costs and higher effectiveness [385]. MRI and CT show similar performance in the presurgical evaluation of pancreatic cancer [386,387]. Preoperative MRI is instrumental to detect the stage and resectability of pancreatic cancer [388] and preoperative gadoxetic acid-enhanced liver MRI has a high diagnostic performance in detecting liver metastasis from pancreatic ductal adenocarcinoma [389,390]. Gadolinium-enhanced MRI with DWI detected synchronous liver metastases [391], whereas [F-18]-FDG PET/MRI provides an imaging tool to improve the staging of pancreatic cancer and for the identification of Sister Mary Joseph nodules [392]. However, the addition of DWI to conventional MRI does not facilitate the differentiation of pancreatic cancer from chronic pancreatitis [393], whereas MRI can differentiate pancreatic carcinoma from chronic pancreatitis successfully when including Gd-enhanced T1-weighted 3D-GE sequences [394]. MRI-guided celiac plexus neurolysis is an effective and minimally invasive procedure for the palliative pain management of pancreatic cancer [395], whereas dynamic susceptibility contrast MRI (DSC-MRI) may predict early progression in patients with advanced pancreatic cancer that are undergoing chemotherapy [396]. MRI has been used to monitor radiofrequency heat (RFH)-enhanced chemotherapy in pancreatic cancers for the efficient treatment of pancreatic malignancies using MRI/RFH-integrated local chemotherapy [397].

#### 4.1.8. Hepatocellular Carcinoma

Primary liver cancer is the second most common cause of cancer mortality worldwide and the sixth most common cancer overall [398]. MRI is superior to CT in sensitivity, specificity, and accuracy [399] and can be used to determine the differential diagnosis [400,401,402], variant analysis [403], arterial phase hyperenhancement [404], small precursor and recurrent lesions [405,406], liver perfusion [407], histological grade, microvascular invasion status, local and systemic therapeutic responses, prognosis [408,409], and as a preoperative marker [410] in hepatocellular carcinoma patients. PET/MRI imaging is also used for the diagnosis of patients with hepatocellular carcinoma [411,412,413], whereas multi-phasic MRI staging was found to be more accurate than the straight hepatocellular carcinoma-grading approach [414]. Texture analysis that was based on gadolinium-ethoxybenzyl-diethylenetriamine penta-acetic acid (GdEOB-DTPA)-enhanced MRI is used for early prediction of therapeutic outcome in intermediate hepatocellular carcinoma [415], identification of vessels encapsulating tumor clusters-positive hepatocellular carcinoma [416], has a higher diagnostic rate and a better diagnostic value in small hepatocellular carcinoma [417], and for the detection of capsule appearance in patients with hepatocellular carcinoma [404] and liver cirrhosis [418,419]. DCE-MRI is used in the prediction of staging B or C hepatocellular carcinoma [415] and for the quantification of perfusion metrics [420] with a superior modality for diagnosis compared with dynamic contrast-enhanced CT-scan [421]. DWI-MRI is the gold standard in detecting recurrent lesions [405], monitoring response to therapy, predicting response, assessing prognosis, and distinguishing tumor recurrence from the treatment effect [422]; however, DWI adds little value to MRI in target delineation [423].

#### 4.1.9. Gallbladder Carcinoma

The most common cancer of the biliary system is gallbladder carcinoma [424]. MRI is a useful imaging tool for the staging, diagnosis, and evaluation of the treatment response and provides superior soft-tissue characterization of the gallbladder and biliary tree [425,426]. DWI is the preferred imaging technique for discriminating benign from malignant disease in gallbladder cancer [427,428].

#### 4.1.10. Renal Cancer

Renal cancer (that is neoplasia of the kidney, renal pelvis, or ureter (ICD-9 189 and ICD-10 C64-C66)) is the seventh most common malignancy in men [429]. There are three main risk factors for cancer of the kidney: age, smoking, and obesity [430]. Renal carcinoma is often first detected and characterized with imaging, with CT and MRI being the most common modalities that are used for diagnosis, staging, and surveillance [431,432,433]. MRI differentiates papillary renal cell carcinoma from other renal masses [434], whereas MRI and normalized ADC has utility in differentiating central renal cell carcinoma from renal pelvic urothelial carcinoma [435]. DCE MRI allows an estimation of the grading of renal cell carcinoma [436] and along with DWI MRI and multiphase contrast-enhanced MRI (MCE-MRI) contributes with prognostic information, even at baseline scans, by predicting the tumor response early after initiating therapy [437].

#### 4.1.11. Bladder Cancer

Bladder cancer is the fourth most common cancer worldwide [438]. MRI is effective in bladder cancer staging as well as differentiating superficial from invasive tumors and organ-confined from non-organ-confined tumors [439,440,441,442,443,444,445,446,447]. MRI has shown potential for the detection of muscle invasion [448]. Mp-MRI has been a useful modality for the T staging of bladder cancer for clinical and research applications [449,450], whereas DCE-MRI provides response biomarkers in clinical trials in subjects with primary bladder cancer [451]. For bladder cancer patients, diagnostics that are based on the use of hybrid systems incorporating both MRI scanning capabilities with the linear accelerator offers a number of potential advantages [452], whereas in bladder cancer patients that are undergoing cystectomy, DWI is used in the detection of metastatic pelvic lymph nodes [453] and in the preoperative T staging of urinary bladder cancer [454].

#### 4.1.12. Ovarian Cancer

Ovarian cancer is the most lethal gynecologic malignancy; it accounts for 2.5% of all malignancies among females but 5% of female cancer deaths because of low survival rates that are largely driven by late-stage diagnoses [455]. An MRI of ovarian cancer has been instrumental to differentiate metastatic ovarian tumors from primary epithelial ovarian cancers [456]. Functional MRI techniques such as tumor-selective molecular imaging (TSMI), DW-MRI, and DCE-MRI are under evaluation as possible predictive and prognostic biomarkers in the context of ovarian malignancy and in routine clinical practice [457,458,459,460,461]. Contrasting results are reported about the role of PET/MRI in ovarian cancer [462]. MRI was found to be more sensitive than PET/CT for detecting local pelvic recurrence and peritoneal lesions of recurrent ovarian tumors [463,464], although PET/CT had a higher specificity than pelvic MRI for diagnosis of malignant ovarian tumors [465,466,467]. A consensus process in the creation of a standardized lexicon for ovarian and adnexal lesions for MRI and the resultant lexicon has been recently published [468].

#### 4.1.13. Cervical Uterine Cancer

Cervical uterine cancer is the leading cause of morbidity and mortality in women in developing countries and is known to be related to human papillomavirus [469]. Pelvic MRI is the reference examination for the evaluation of cervical cancers, allowing the accurate evaluation of tumor size, parametrial extension, and lymph node metastasis, which are essential factors for therapeutic management [470,471,472,473]. MRI, CT/MRI, and PET/MRI have been used for cervical cancer staging and lymph node metastasis [474,475], and PET/MRI was found to possess a higher diagnostic sensitivity [476], specificity [477], and accuracy [478,479], also during pregnancy [480], being helpful in clinical diagnosis [481], prediction [482], and treatment [483,484]. MRI diagnosis is an auxiliary method for cervical cancer treatment when used in combination with tumor markers (e.g., squamous cell carcinoma antigen) [485] and for the management of women with early cervical cancer [486]. DWI-MRI and ADC are used as a non-invasive imaging methods for characterizing the fraction of collagen I-positive tissue across different tumor models of uterine cervical cancer [487], for the pathological grade of tumor [488], for the differentiation between metastatic and non-metastatic pelvic lymph nodes [489], and between normal and cancerous tissue in the uterine cervix [490]. Another noninvasive technique that is used to assess tumor vascular oxygenation at 3 T in cervical cancer staging is blood oxygenation level-dependent contrast MRI [491].

#### 4.1.14. Endometrial Cancer

Endometrial cancer is the fourth most common malignancy in women and the most common gynecological malignancy in the developed world after lung, colorectal, and breast cancer [492]. MRI is recommended for the initial staging and report of endometrial cancer [493,494,495,496] and preoperative pelvic MRI is a moderately sensitive and specific method of identifying invasion to the outer half of myometrium in endometrial cancer [497,498,499]. MRI has a high specificity and negative predictive value in endometrial cancer staging [500]; however, its accuracy in detecting myoinvasion is limited [501]. MRI with DWI and DCE sequences can help establish a correct diagnosis [502,503], while 3.0 T multimodal MRI is an important imaging tool for preoperative endometrial cancer staging [504]. MRI quantitative assessments such as tumor area ratio (TAR), tumor volume ratio (TVR), MRI-based whole-tumor radiomic signatures, and ADC were found to improve the accuracy of preoperative staging, helping in the risk stratification of endometrial cancer [505,506,507]. The combination of MRI and immunohistochemical examination is a powerful tool for preoperative risk stratification to assist in clinical decision-making for endometrial cancer patients [508]. [F-18]-FDG PET/MRI is a valid imaging technique in patients with endometrial cancer, both in staging and restaging as an alternative diagnostic strategy to conventional imaging modalities, also considering the limited radiation exposure [509,510,511], whereas integrated PET/MRI successfully assesses the lymph node metastasis and myometrial invasion in patients with endometrial cancer [512,513]. MRI-guided intensity-modulated radiation therapy has been used for locally recurrent endometrial cancer after resection [514].

#### 4.1.15. Prostate Cancer

Prostate cancer is the most common cancer for males, and it is estimated that 15% men are predicted to develop prostate cancer over their lifetime [515]. The application of MRI has been successfully used for its sensitivity in detecting clinically significant cancer and the ability to localize the tumor within the prostate gland [516,517] by using Mp-MRI [518,519,520,521,522] and in the hybrid PET/MRI [523,524,525]. In addition to the fusion strategy, biopsies with MRI targets play an important role in the assessment of patients with a previous negative prostate biopsy [518].

#### 4.1.16. Testicular Cancer

In men between 15 and 49 years-old, the most common nonhematologic malignancy is testicular cancer [526]. It has excellent cure rates; however, poor guideline adherence can lead to inappropriate management with a detrimental effect on outcomes [527]. MRI is successful in the diagnosis of testicular germ cell cancer [528,529], particularly when directed towards the retroperitoneum and pelvis only [530]. Functional information that is based on DWI and DCE MRI data improve testicular mass lesion characterization [531,532] and can be used to characterize small, solid testicular tumors [533] and for the follow-up of testicular cancer patients [534], independently of the examiner [535]. Mp-MRI can potentially differentiate benign stromal tumors from malignant testicular neoplasms, which can help to avoid radical orchiectomy [536,537]. MRI can be used as an alternative to CT to reduce radiation exposure [538,539] and is a valuable diagnostic aid in the preoperative localization of ectopic testes in cryptorchidism or if findings are equivocal [540].

#### 4.1.17. Colorectal Cancer

Colorectal cancer is a common cancer and a common cause of death. There is evidence that an important proportion of colorectal cancer patients remain untreated [541]. Pelvic MRI is used for the local of T and N staging of rectal cancer and has the advantage of improved patient comfort [542], improved reproducibility and accuracy [543], reduced care costs [544], and for completeness and better understanding of related pelvic anatomy [545,546,547]. The use of DWI with ADC value in addition to conventional MRI yields better diagnostic accuracy than using conventional MRI alone in detection, correlation with tumor histologic grade, and the initial staging in patients with locally advanced colorectal cancer [548,549]; however, DW-MRI is inferior to [F-18]-FDG-PET for the detection of primary lesions but superior for the detection of lymph node metastases [550]. MRI shows moderate sensitivity and good specificity for the detection of extramural venous invasion (EMVI) in colorectal cancer [551,552], while 3D colorectal MRI gives better and accurate segmentation results than 3D fully convolutional neural networks alone [553]. For lymph node metastasis of colorectal cancer, the sensitivity and specificity of preoperative diagnosis by diffusion (D-MRI) is higher if the node is hyperintense and more than 9 mm in diameter [554], whereas high-b-value DWI-MRI has a high sensitivity and specificity to detect colorectal adenocarcinoma [555]. Recent developments and emerging technologies in CT and MRI are changing the management of colorectal cancer patients in many clinical scenarios [556]. MRI is more accurate than CT [557] and MDCT for the evaluation of liver metastases [558], whereas for patients with colorectal cancer, PET/MRI may aid in the selection of more appropriate treatment strategies [559]. Whole-body MRI (WB-MRI) is a radiation-free alternative to standard sequential algorithms of staging and follow-up of colorectal cancer [560].

### 4.2. Magnetic Fields and Cancer Treatment

Radiation therapy, chemotherapy, and immunotherapy, alone or in combination with therapies such as photothermal therapy, photodynamic therapy, hyperthermia, and radiotherapy have been proposed in the recent literature [561]. In the next section, the use of magnetic nanoparticles for the delivery of anticancer agents and magnetomechanical tools and in hyperthermia will be discussed.

#### 4.2.1. Delivery of Anticancer Agents via Magnetic Carrier Particles

An exciting new prospect in treating cancer is the delivery of anticancer agents via magnetic carrier particles, which are used as a "carrier system" for a variety of anticancer agents [562,563,564,565]. For instance, by using an external MF, it is possible to guide magnetic iron oxide nanoparticles (MIONs) to their target. This is the principle behind the development of superparamagnetic iron oxide nanoparticles (SPIONs) as novel drug delivery vehicles [566]. Palmitoyl chitosan that is co-encapsulated with SPIONs and the anticancer drug paclitaxel via the nanoprecipitation process increased the amount of drug in cancer cells [567], whereas doxorubicin (Dox)-conjugated heparin was used with the SPION technology for targeted anticancer drug delivery [568]. In MCF-7 breast cancer cells, Dox was rapidly internalized and exhibited higher toxicity than treatments with Dox alone when it was assembled in magnetic nanoparticle-supported lipid bilayers [569] and with Dox-loaded polymer@Au/Fe_3_O_4_ core/shell nanoparticles in simulated cancerous environments [570]. Bioactive molecules such as curcumin can be loaded in magnetic silk fibroin core-shell nanoparticles to enhance cytotoxicity and cellular uptake in the human Caucasian breast adenocarcinoma cell lines with superior biomedical applications due to their size range, which is particularly desired for cell internalization [571].

It has to be considered that since the magnetic force decreases rapidly with the distance from the magnets, the targeting of tumors that are situated at large distances from the surface of the human body might be difficult. Therefore, the delivery of anticancer agents via magnetic carrier particles appears to be more suitable for treating sub-surface cancers within the human body [572]. Nevertheless, several new magnetic nanoparticles have been designed and evaluated for cancer treatment, offering the ability to deliver drugs efficiently [573,574,575].

Despite the increasing body of evidence supporting promising results, there are some drawbacks that are related to magnetic nanoparticles (MNP) use in drug delivery, such as the difficulty in maintaining the therapeutic action in three dimensions inside the human body, the limitation to maintain efficacy in the target organ once the MF is removed from outside, and the limited effective incorporation of magnetic iron oxide nanoparticles into biomedical systems [576,577].

#### 4.2.2. Magnetomechanical Methods for Cancer Therapy

Magnetomechanical therapy is one of the most prospective directions in tumor microsurgery based on a physical nanostructure that is able to transform the magnetic moment to mechanical torque and a ligand molecule that allows the scalpel to precisely target tumor cells [578]. Nano-magnetomechanical activation (NMMA) of the MNPs is used to localize and apply force to target biomolecular structures as transport vesicles, cell organelles, enzyme molecules, etc., without significant heating [579]. Nanospinners can exert mechanical forces under a rotating magnetic field at 15 Hz and 40 mT to target the mitochondria of cancer cells [580]. Iron nanowires that are functionalized with anti-CD44 antibodies have been used in a combination therapy that included magnetomechanical and photothermal treatments on colon cancer cells [581]. Hedgehog-like microspheres that were composed of needle-like MNP with carbon and gold double shells seriously damaged cancer cells and strongly inhibited tumor growth through mechanical force [582]. Magnetic disks are a new generation of MNP with outstanding properties to face biomedical challenges in cancer treatment microsurgery [578] and the investigation toward the most efficient magnetomechanical actuator to destroy cancer cells has been recently reviewed [583].

#### 4.2.3. Magnetic Hyperthermia Ad Cancer Treatment

Magnetic hyperthermia treatment (MHT) utilizes heat that is generated by MNPs under an alternating MF to selectively kill tumor cells [584,585,586]. When exposed to an alternating magnetic field (AMF), MNPs can generate heat via hysteresis loss (large multi-domain MNPs) or through Neel- and Brownian relaxation losses (typically small, single-core MNPs) [587]. The efficiency of MHT depends on the size, concentration, solution viscosity, and composition of MNPs as well as the strength and frequency of the MF [588]. 

Several materials are used to prepare MNP for MHT. Ferrimagnetic glass-ceramic have been successfully used as thermo-seeds for a hyperthermic treatment of carcinoma cells in Sprague–Dawley rats [589], whereas magnetite cationic liposomes where used to generate hyperthermia on local tumors and lung metastases in a mouse model of osteosarcoma [590]. Spinel ferrite nanoparticles were successfully synthesized and used for MHT [591], whereas MIONs (such as crack-free ferrimagnetic maghemite, γ-Fe_2_O_3_) may be useful for the in situ hyperthermic treatment of cancer [592,593,594,595]. SPIONs have been increasingly studied for their excellent MHT applications [596,597], whereas lanthanum-strontium manganite particles that were embellished with gold nanoparticles were found to be suitable for the treatment of deeper tumors [598].

The limitations and advantages to more effective clinical use of MNP-based thermometry to achieve greater impact on clinical translation of MNH have been recently reported [599]. Despite the wide use of MHT in clinical application, the technology suffers from inadequate and uneven heating due to low and heterogeneous concentrations of MNPs within the target tumor [600,601]. It has been calculated that, to achieve sufficient hyperthermia in targeted tumors, a high concentration of MNP is required [602] and often the particle concentration is below that which is needed to induce sufficient heating of tissue, thus lowering the therapeutic effects of MHT [603].

The use of MNPs in “traditional” biomedical applications that are related to cancer theranostics, such as drug delivery, hyperthermia, MRI, micro nuclear magnetic resonance, and surface-enhanced Raman spectroscopic detection technology has been demonstrated [604,605,606,607,608,609], including the development of next-generation high-performance theranostic agents that are based on MNP assemblies [610]. Figure 6 summarizes the use of MFs in cancer theranostics.

## 5. Conclusions

The potential health effects of man-made EMF have been a topic of scientific interest since the late 1800s and have received particular attention during the last 40 years. Since the first studies suggesting a relationship between MF and childhood cancer [29], the scientific community has evaluated the possible mechanisms for the effects of MFs on biological systems. Epidemiological studies are often controversial and sometimes misleading. Nevertheless, there is a consensus on the positive relationship between residential/domestic exposure to ELF EMF and the occurrence of brain cancer, whereas contrasting results require more experimentation to assess the influence of occupational exposure to MFs on brain cancer. The epidemiology of leukemia as related to ELF EMF in adults is controversial in both residential/domestic and occupational exposure. For children, leukemia is not associated to occupational exposure, whereas a growing body of evidence indicates a correlation between residential/domestic exposure to ELF EMF and childhood leukemia. Breast cancer has been related to ELF EMF exposure more in residential/domestic epidemiological studies than in occupational, but the melatonin hypothesis, although recently revisited, finds little consensus. When studied at the cellular and in vitro level, MFs exert their effect on both human and animal (rat and mice, mainly) cells when used at a high intensity and for a long time. The common response is the production of ROS, which trigger a cascade of other cellular responses which might be the direct consequence of MF exposure. The use of MF is instrumental for the diagnosis and therapy (theranostic) of cancer. MRI is instrumental for the precise diagnosis of different cancers, whereas MNPs open the new era of nanomedicine, allowing (i) the smart delivery of anticancer drugs, (ii) nanosurgery through their magnetomechanic properties, and (iii) fighting the cancer cells in situ by exploiting their capability to generate heat (hyperthermia) via hysteresis loss or through Neel- and Brownian relaxation losses.

Figure 7 summarizes the effects of MFs on cancer that were discussed in this review.

Although humans do not perceive the presence or changes of MFs, variations in MF intensity and inclination exert biological effects, with the greatest effects observed at the cellular and subcellular level. The basic response to MF relies on ROS-production with RPM playing a potential role in magneto-perception. Scientists do not generally agree that there is a cause-effect relationship between exposure to MF and cancer, also because of the difficulty in obtaining reproducible effects that are consistent with the hypothesis that MF may cause or promote cancer. MFs might not be the direct cause of cancer but may contribute to ROS-production and generate oxidative stress through RPM [611], which could trigger or enhance the expression of oncogenes [612]. Large-scale epidemiological studies are needed to help resolve these issues along with in depth studies on the relationship between magnetoreception, ROS-generation, and cancer.

## Figures and Tables

**Figure 1 ijms-23-01339-f001:**
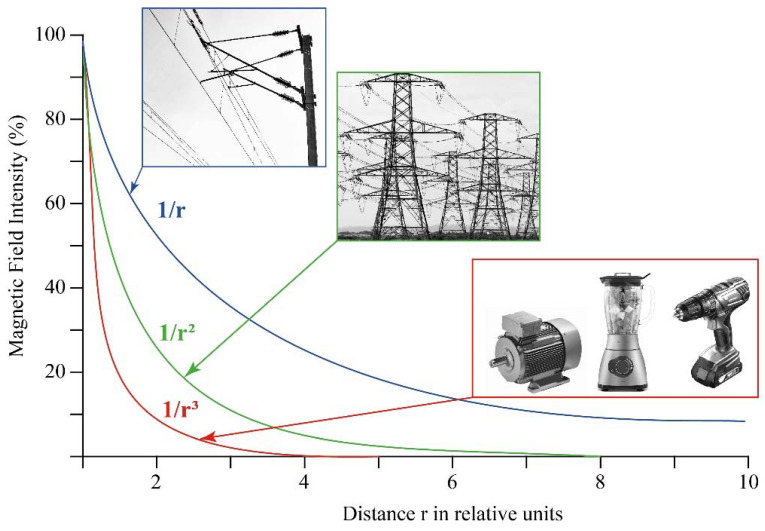
The magnetic field intensity decreases with growing distance from the field source.

**Figure 2 ijms-23-01339-f002:**
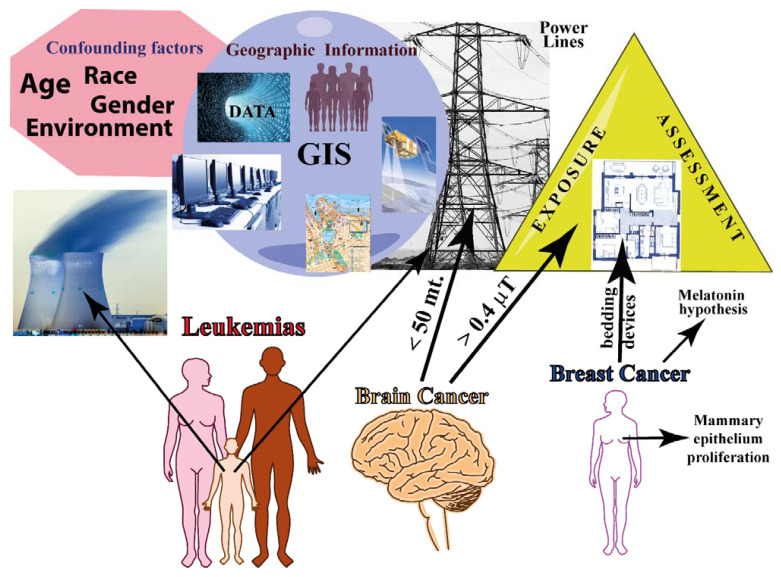
Summary of epidemiology of residential/domestic exposure to MF. The major confounding factors in epidemiological studies are shown along with the main geographical information that is based on GIS (data management, hardware/software, topography, remote detection, and population demographics). Exposure assessment needs to be evaluated both outside and inside the residence. The three major cancers are represented: leukemia affects mainly childhood; brain cancer increases with decreasing distances from EMF sources; and breast cancer is associated with mammary epithelium proliferation and with exposure to bedding devices.

**Figure 3 ijms-23-01339-f003:**
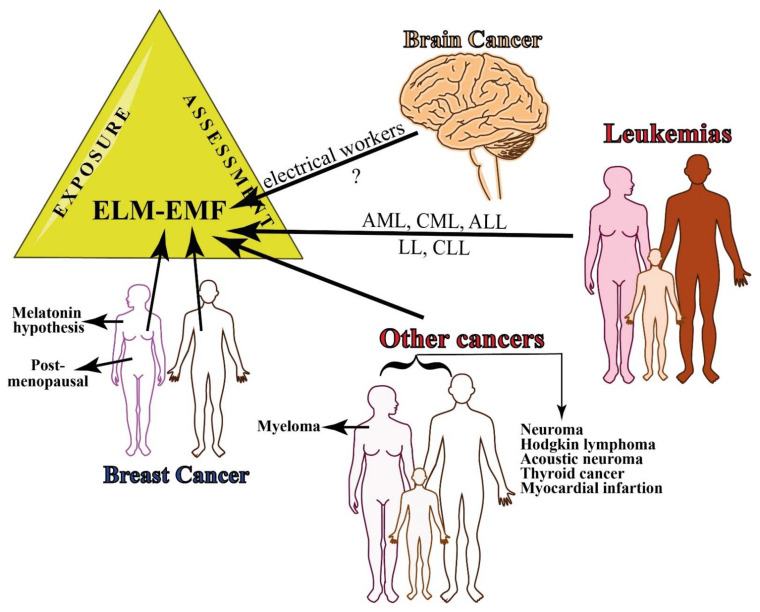
Summary of epidemiology of occupational exposure to MF. There is still little evidence on the relationship between occupational exposure to EMF and brain cancer, whereas several leukemias have been associated with continuous exposure to ELF-EMFs. Breast cancer occurs both in women and men and the risk increases in men that are exposed to 0.6 T. Other cancers that are associated with MF exposure include myeloma in women and several other types of cancer in both women and men.

**Figure 4 ijms-23-01339-f004:**
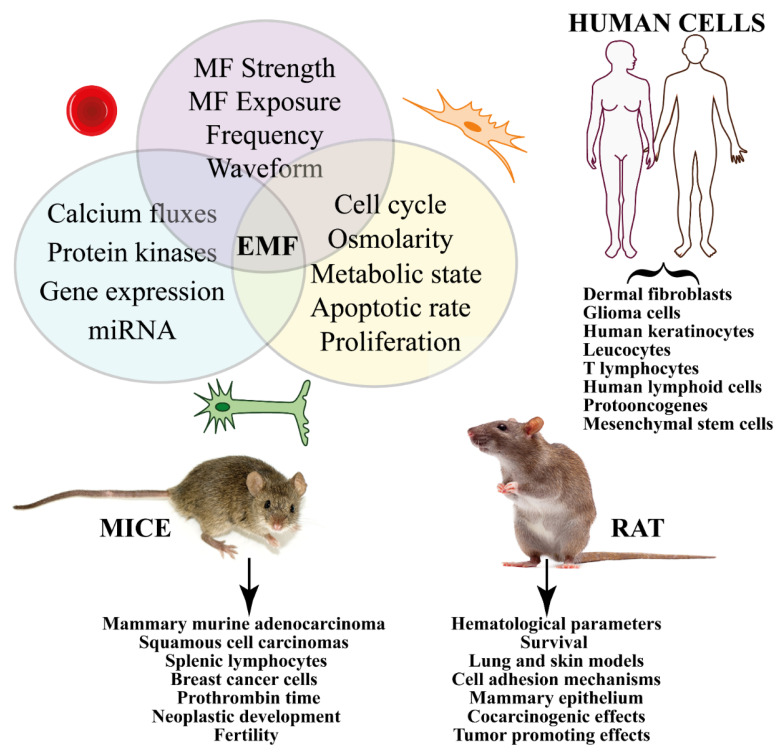
Summary of the in vivo and in vitro effects of MFs on cancer from studies on human and animal cells. Different treatments (e.g., strength, duration, frequency, etc.) induce signal transduction pathways that eventually trigger gene expression. In vitro studies show significant effects on cell cycle, proliferation, and apoptosis. Human cells have been used to evaluate the effect of MF on several cancer types, whereas animal experimentation has been focused on mice and rats.

**Figure 5 ijms-23-01339-f005:**
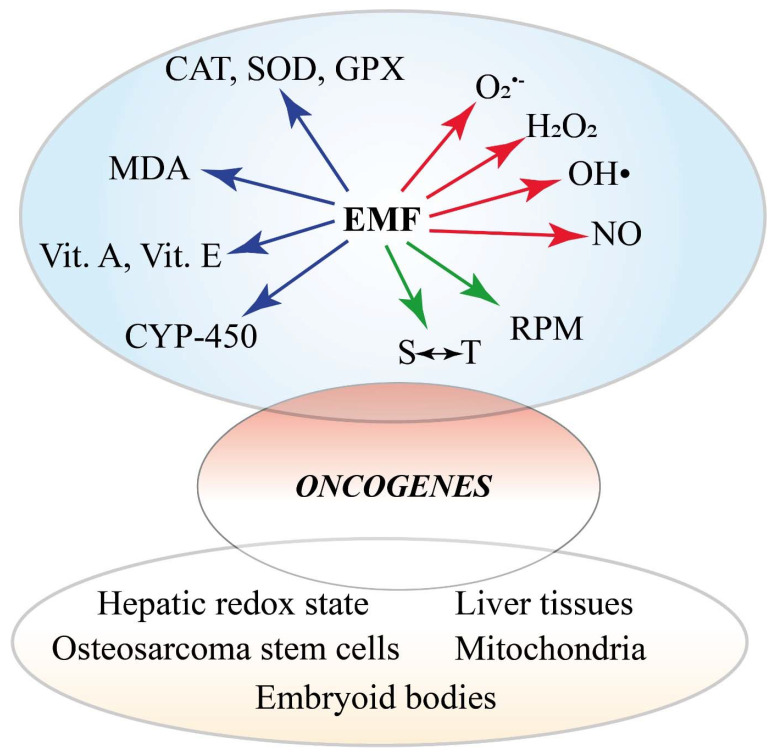
Summary of the involvement of ROS and RNS in the cellular and organismic responses to MFs. There are three major effects of varying MF the are reported. MFs alter the redox status of the cell by affecting the activity and gene expression of ROS-scavenging systems, including CAT, SOD, GPX, vitamins, and monooxygenases. Membrane degradation is evidenced by MDA detection. On the other hand, MFs trigger the production of ROS and NOS and early events involving the radical pair mechanism and spin-chemical effects. The altered oxidative status eventually induces the expression of oncogenes. The alteration of the oxidative status is also evident at the subcellular, cellular, tissue, and organ level.

**Figure 6 ijms-23-01339-f006:**
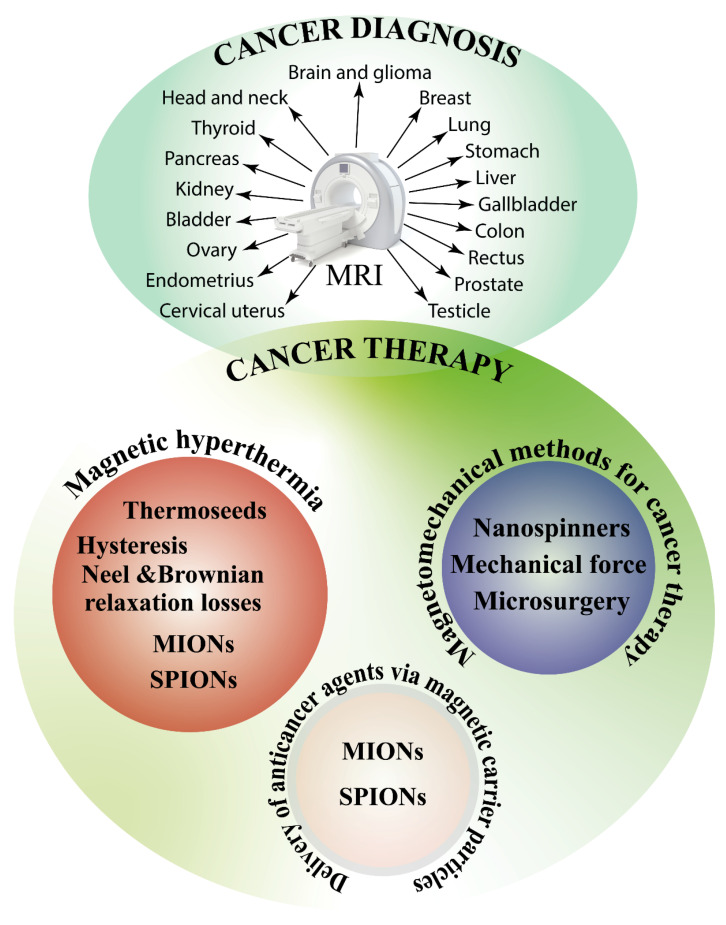
MFs and cancer theranostics. Theranostics combines the terms diagnostics and therapy. MF are used in MRI for the diagnosis of several cancers, whereas the use of magnetic nanoparticles for cancer therapy encompasses three major areas: magnetic hyperthermia that is aimed to kill cancer cells with heat; drug delivery by the use of SPIONs and MIONs; and the exploitation of MNP mechanical forces for application in nano and microsurgery.

**Figure 7 ijms-23-01339-f007:**
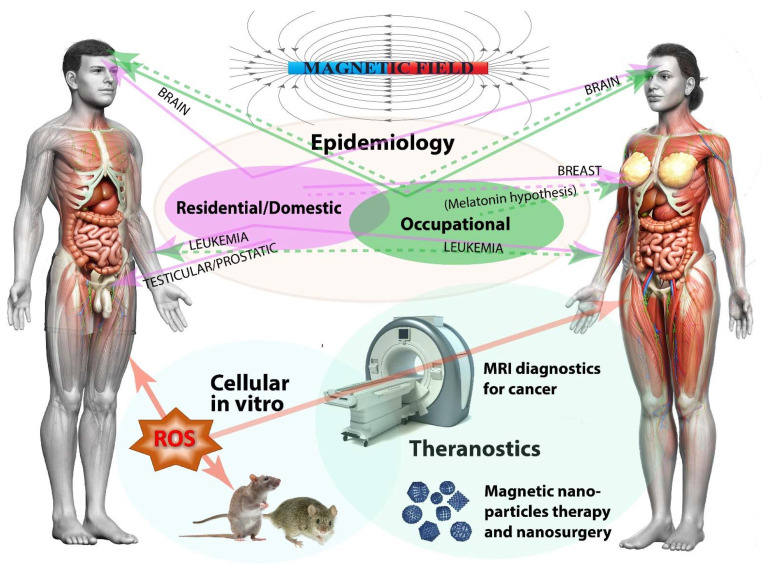
The effects of MFs on cancer. Humans are exposed to a complex mix of man-made electric and magnetic fields at many different frequencies both at home and at work. Epidemiological studies indicate that there is a positive relationship (solid lines) between residential/domestic and occupational exposure to ELF EMF and brain cancer, although some other studies indicate that there is no relationship (dotted lines). Breast cancer appears to be more related to residential/domestic exposure than occupational and in both epidemiological surveys, the so-called “melatonin hypothesis” finds weak evidence. Testicular/prostatic cancer is associated with residential/domestic exposure, as is leukemia, which is mostly associated (particularly in children) with the close proximity to ELF EMF. The cellular and in vitro studies on both animal (mainly rat and mice) and human cells indicate the role of ROS-generation as a consequence of exposure to different MF intensity and timing, suggesting also a magnetoreception mechanism that is based on RPs. Finally, MFs can be used for theranostic applications; MRI is instrumental for the diagnosis of several cancers, whereas the use of MNP allows the treatment of cancer by nanomedical applications for the precise delivery of anticancer drugs, nanosurgery by magnetomechanic methods, and the selective killing of cancer cells by magnetic hyperthermia.

**Table 1 ijms-23-01339-t001:** Types of magnetic fields.

Type of Radiation	Type of Field	Frequency	Wavelength	Use	Examples	Effect
NI	SMF	0 Hz	N.A.		GMF, permanent magnets, transmission lines, HVDC lines, batteries, between objects with different electrical charges, MRI	Action of force
NI	AMF	0.3 Hz 3 Hz 16 2/3 Hz 50 Hz 300 Hz 3 kHz 30 kHz	10^6^ km 100,000 km 18,000 km 6000 km 1000 km 100 km 10 km	Low Frequency traction current and three phase alternating current	Technical appliances such as power lines, wiring and household appliances such as appliances for heating (e.g., electric cooker, electric heating, washing machine, electric water heater, iron), appliances with a transformer or magnetic coils (e.g., radio clock, low-voltage halogen lamps, television set, WiFi) and appliances with an electric motor (e.g., vacuum cleaner, drill, hand blender, hair dryer, electric cars)	Stimulation/irritation
NI	AMF	300 kHz 3 MHz 30 MHz 300 MHz 3 GHz 30 GHz 300 GHz 3 THz	1 km 100 m 10 m 1 m 10 mm 10 mm 1 mm 100 µm	Radio frequency. Radio/television, microwaves, terahertz waves	Induction cookers and electronic article surveillance systems in stores, as well as many industrial and medical applications, PC monitors, mobile phone, microwave ovens, radar stations. Broadcasting frequencies (short wave, AM, and FM radio), digital television (digital video broadcasting-terrestrial, DVBT) and digital radio (digital audio broadcasting, DAB). Wireless local area networks (WiFi, WLAN), cordless telephones, Bluetooth devices, baby monitors, electronic article surveillance systems and RFID (radio frequency range), radar systems, radio relay systems, satellite TV and satellite Internet, radio solutions for stationary Internet	Thermal effect
NI	AMF	30 THz 300 THz	10 µm 1 µm	Infrared	Bulb lamps, heaters, body scanners for security control	Thermal effect
NI	AMF	380 THz 789 THz	780 nm 380 nm	Visible Light		
Ionizing	AMF	3 × 10^15^ Hz 3 × 10^16^ Hz 3 × 10^17^ Hz 3 × 10^18^ Hz 3 × 10^19^ Hz 3 × 10^20^ Hz	100 nm 10 nm 1 nm 100 pm 10 pm 1 pm	UV-light, X-rays, gamma rays	Nuclear power plants, X-ray machines, radioactive material.	Ionization

## Data Availability

Appendix A contains the EndNote Library used for this review.

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
