# Peer review of "Magnetic Fields and Cancer: Epidemiology, Cellular Biology, and Theranostics"

_ijms, 2022, doi:10.3390/ijms23031339_

Round 1

Reviewer 1 Report

The author reviewed “The relationship between magnetic fields and cancer in terms of epidemiology, cellular biology, and theranostics”. Overall, the manuscript covers from definition to examples of magnetic fields and explained the correlation between magnetic fields and cancer in various aspects such as epidemiological studies, cellular biology studies, and theranostics. The manuscript is well organized and comprehensively described. Also, it is fully covered with a bunch of appropriate and adequate references. The one suggestion is to include more figures and tables to increase readability instead of reduces the amount of texts, they will help readers discover and understand easily what they are curious about. 

Author Response

Thanks a lot for the useful suggestions. I have inserted different figures that summarize the topics described. I decided to leave the Supplementary tables as supplementary in order not to expand too much the review. Readers can easily refer to these tables.

Reviewer 2 Report

It is a well-written review article. I recommend accepting in the present form. 

Author Response

Thank you very much

Reviewer 3 Report

The manuscript presents a review on the influence of electromagnetic fields, particularly of magnetic fields (static and frequency dependent), on the inducement, detection and treatment of several forms of cancer. The first part is dedicated to the definitions of the concepts, to the sources of electromagnetic fields and to their effect on the human body. The review then proceeds to discuss the literature studies and results on the correlations (or lack of them) between the magnetic fields and several cancers, on the effects of magnetic fields on cancer cells and on the use of magnetic fields in the detection and in the treatment of cancers. The use of magnetic nanoparticles for drug carrier, for magnetic hyperthermia and for magnetomechanical treatment is also discussed. The review is very detailed, comprehensive and with abundant, and recent, references and only needs minor revisions. I have the following comments:

- On page 2 it is written “Whereas the total field intensity, which is directed towards the center of the planet”. At the earth surface the earth magnetic field is approximately similar to a dipole magnetic field. It only points towards the center near the poles (closing flux lines). It would be better to clarify the sentence.

- On page 3 it is written that “whereas in SMFs the polarity is almost unchanged”. What do the author mean by “almost unchanged” ? A static magnetic field (SMF), as the name refers, never changes.

- On page 5 it is written that “Radio frequency (10 MHz–300 GHz) include a range of "broadcasting frequencies" (between 30 kHz and 300 MHz; wavelengths from 10 km to 1 m)”. However, the 30 kHz to 10 MHz broadcasting frequency range is not in the defined radio frequency range. The second interval (30 kHz-300 MHz) is not completely included in the first (10 MHz-300 MHz). The text should be more clear on the relations between the different types of frequency regions.

- On page 5 it is written “The following range of 789 THz to 384 THz (380 nm to 780 nm) is referred to as visible light”. It would be better to put increasing frequencies (decreasing wavelengths) as was done in the previous part of the text, otherwise it is confusing for the reader. In this respect, where it is written “This is preceded by the ranges of ultraviolet radiation and the ionizing radiation” should be written “This is succeeded by the ranges of ultraviolet radiation and the ionizing radiation”, for clarity.

- On page 22 it is written “Dox was rapidly internalized and exhibited higher toxicity than treatments with Dox alone”. Does this toxicity refer to the human body or only to the cancer cells ?

- Some mistakes: “are associated low frequency electromagnetic fields” should be “are associated with low frequency electromagnetic fields” on page 3, “magnetic flux density lower down to” should be “magnetic flux density lowers down to” on page 3, “residential fields considered limited” should be “residential fields are considered limited” on page 8, “0.3/0.4 muT” on page 8 (mu should be the greek letter), “shoe” should be “show” on page 10, “dysregulated” should be “deregulated” on page 15, “and offers provided” should be “and provides” on page 18, “finds little consent” should be “finds little consensus” on page 24.

Author Response

- On page 2 it is written “Whereas the total field intensity, which is directed towards the center of the planet”. At the earth surface the earth magnetic field is approximately similar to a dipole magnetic field. It only points towards the center near the poles (closing flux lines). It would be better to clarify the sentence.

R: Thanks a lot for this remark, the sentence has been clarified

- On page 3 it is written that “whereas in SMFs the polarity is almost unchanged”. What do the author mean by “almost unchanged” ? A static magnetic field (SMF), as the name refers, never changes.

R: Thanks for noticing this, the term "almost" has been deleted

- On page 5 it is written that “Radio frequency (10 MHz–300 GHz) include a range of "broadcasting frequencies" (between 30 kHz and 300 MHz; wavelengths from 10 km to 1 m)”. However, the 30 kHz to 10 MHz broadcasting frequency range is not in the defined radio frequency range. The second interval (30 kHz-300 MHz) is not completely included in the first (10 MHz-300 MHz). The text should be more clear on the relations between the different types of frequency regions.

R: Thanks for noticing this, the range has been edited

- On page 5 it is written “The following range of 789 THz to 384 THz (380 nm to 780 nm) is referred to as visible light”. It would be better to put increasing frequencies (decreasing wavelengths) as was done in the previous part of the text, otherwise it is confusing for the reader. In this respect, where it is written “This is preceded by the ranges of ultraviolet radiation and the ionizing radiation” should be written “This is succeeded by the ranges of ultraviolet radiation and the ionizing radiation”, for clarity.

R: Thanks for noticing this, changes have been made according to your suggestions

- On page 22 it is written “Dox was rapidly internalized and exhibited higher toxicity than treatments with Dox alone”. Does this toxicity refer to the human body or only to the cancer cells ?

R: The sentence has been clarified and specified

- Some mistakes: “are associated low frequency electromagnetic fields” should be “are associated with low frequency electromagnetic fields” on page 3, “magnetic flux density lower down to” should be “magnetic flux density lowers down to” on page 3, “residential fields considered limited” should be “residential fields are considered limited” on page 8, “0.3/0.4 muT” on page 8 (mu should be the greek letter), “shoe” should be “show” on page 10, “dysregulated” should be “deregulated” on page 15, “and offers provided” should be “and provides” on page 18, “finds little consent” should be “finds little consensus” on page 24.

R: All typos have been corrected, thanks a lot for noticing this

Reviewer 4 Report

The manuscript entitled “Magnetic fields and cancer. Epidemiology, cellular biology and theranostics” is this is a review article devoted to the main scientifically based studies of the dependencies of cancer development on exposure to magnetic fields of different frequencies. The review contains structured information on each type of cancer separately. It should be noted a competent approach based on the Bradford Hill criteria, which allows researchers to be as objective as possible when studying such a complex issue. Contradictory data is embedded in the overall concept and conclusions are drawn quite carefully, where possible.
The study can be accepted after minor revision.
Line 139 contains the name of table 1, followed by the table. It is advisable to move the table to the next page in its entirety.
After line 1136, it would also be necessary to transfer the list of abbreviations to a separate page, to simplify perception.
Minor flaws do not change the overall impression of a perfectly written article.

Author Response

Line 139 contains the name of table 1, followed by the table. It is advisable to move the table to the next page in its entirety.

R: Thank you, the Table is in the next page now

After line 1136, it would also be necessary to transfer the list of abbreviations to a separate page, to simplify perception.

R: Thank you, the Abbreviation title is in a new page now

Minor flaws do not change the overall impression of a perfectly written article.

R: Thank you very much